# Review of Tribological Failure Analysis and Lubrication Technology Research of Wind Power Bearings

**DOI:** 10.3390/polym14153041

**Published:** 2022-07-27

**Authors:** Han Peng, Hai Zhang, Linjian Shangguan, Yisa Fan

**Affiliations:** School of Mechanical Engineering, North China University of Water Resources and Electric Power, Zhengzhou 450000, China; penghan@ncwu.edu.cn (H.P.); zhanghai09484@163.com (H.Z.)

**Keywords:** wind power bearings, bearing lubrication, failure analysis, lubrication technologies

## Abstract

Wind power, being a recyclable and renewable resource, makes for a sizable portion of the new energy generation sector. Nonetheless, the wind energy industry is experiencing early failure of important components of wind turbines, with the majority of these issues also involving wind power bearings. Bearing dependability is directly tied to the transmission efficiency and work performance of wind turbines as one of its major components. The majority of wind turbine failures are due to bearings, and the vast majority of bearing failures are due to lubrication. The topic of improving the accuracy and life of wind power bearing motion is becoming increasingly essential as the wind power industry develops rapidly. This study examines the various constructions and types of wind turbines, as well as their bearings. We also examined the most typical causes of friction and lubrication failure. Furthermore, contemporary research on wind turbine bearings has been compiled, which mostly comprises the study and development of lubrication technology and other areas. Finally, a conclusion and outlook on current challenges, as well as future research directions, are offered.

## 1. Introduction

With increasingly severe environmental challenges and energy crises, wind energy, as an efficient-clean resource, is playing an important role in the power generating industry and the transition from new to old energy [1,2]. The wind energy industry has already begun an era of rapid development [3]. Furthermore, statistical data reveal that the wind energy markets in China and the United States have emerged as the primary engines driving global wind power growth [4]. As Figure 1 illustrates, according to the IEA, in 2020, China’s installed onshore wind capacity almost triples the growth of 2019 to nearly 69 GW; similarly, wind energy deployment in the US was twice as high as in 2019; new installed capacity in the EU and the UK remained at 8 GW; while India’s wind capacity fell by 50% to just 1 GW. The global wind power industry is predicted to add 1 TW of installed capacity by 2030, and China’s wind power market will account for 41% of the total new installations worldwide [5,6].

Wind energy generation is a critical endeavor for advancing the circular economy and alleviating resource constraints. Unlike traditional energy sources, which create massive consumption and pollution, wind power, as a technology that converts electricity from renewable resources, produces no pollutants during power generation. As a result, it is extremely important for energy conservation and emission reduction, and it is one of the most effective strategies to fundamentally reduce environmental pollution. Furthermore, the economic and environmental benefits of wind power generation will outweigh traditional power generation methods in the long run, and its energy-saving and emission-reduction effect is exceptional, indicating a new trend in future power development. However, the accelerated growth of the wind energy industry has also brought new challenges and problems [7]. Because of the complex operating conditions, quickly altering temperature, air pressure, wind speed, and load makes the power equipment vulnerable to failure [8,9]. Among them, bearings, as the core components of critical sections, have a significant effect on the transmission system and service behavior, which help to provide smooth rotational motion [10]. Nonetheless, the life cycle of bearings can be greatly affected by unbalanced operating loads and environmental circumstances, which can also increase the MRO (Maintenance, Operation, and Reliability) costs [11,12]. Therefore, it is crucial to implement effective research and analysis of wind turbine bearings to avoid power production losses and parts repair expenses from device failures [13].

Bearing health is crucial to wind turbine operation because numerous failure modes in bearings (such as abrasion, micro-etching, white etched cracks, micro-motion corrosion, and so on) can cause irreversible damage to turbine components. To mitigate such issues, it is critical to identify the root causes of failures and develop potential solutions so as to maximize the expected life of the wind turbine components, reduce maintenance costs, and make wind production less costly, thus reducing non-essential energy consumption. In regards to our work, we aim to conduct the review in two parts.

The common tribological failures of wind turbine bearings are analyzed, and the typical failure modes of bearings are reviewed in detail, specifically in terms of both friction and lubrication. The application of analysis methods such as FTA helps the reader to understand the causes and effects of bearing failures.The research and developments in wind turbine bearings lubrication technology are discussed, with the goal of utilizing cutting-edge research techniques to increase predicted bearing life and lessen current tribological failure issues. The study on lubricants for wind turbine bearings, the state of lubrication method research, and the development of lubrication monitoring systems are all thoroughly discussed. To fill the gap left by the existing lubrication methods used on big wind turbine bearings, certain innovative lubrication approaches applied to conventional mechanical bearings are also summarized.The remainder of the paper is organized as follows. Section 2 collates the composition, classification and operating conditions of wind turbines and their bearings. Section 3 analyzes the typical tribological failure modes and causes of bearings, and the FTA analysis method is used to provide a summary and generalization. Section 4 presents a detailed overview of recent research on wind power bearings lubrication technology and summarizes the advantages and disadvantages of existing technologies and methods. Finally, Section 5 gives conclusions and provides an outlook on future research trends, which provides a direction for the future development of green lubrication of wind turbine bearings.

## 2. Wind Turbine and Bearings Composition and Classification

The major types of wind turbine bearings relate to the central units in the spindle, pitch, yaw, gearbox, and generator systems used in wind turbines, corresponding to spindle, pitch, yaw, gearbox, and generator bearings, respectively. Statistically, the number of bearings in a typical onshore wind turbine is about 26 sets [14,15,16].

### 2.1. Composition and Classification of Wind Turbine

Wind turbine is a machinery that facilitates the transformation of wind power into electrical energy. The main sections include wind machine (transfer wind energy to mechanical energy) and generator (shifting mechanical energy to electrical energy) [17]. There are many categories of wind turbines which, based on the position of main shaft relatively to the ground, the sort of generator, blade force patterns, and operation modes, etc., can be classified into different types [18,19,20]. Common types of wind turbines are shown in Table 1.

Currently there are various types of wind turbines, but the most commonly used and technically developed is horizontal axis with three blades [21,22], as shown in Figure 2.

The three-bladed horizontal axis with the gearbox, generator, and rotor integrated in the cabin can provide higher electrical generation and power output. The blades are installed in the counter-wind position of the tower and nacelle. The nacelle is generally fitted with an anemometer to measure wind speed and direction. The yaw system is positioned in the space between the nacelle and tower for driving the nacelle, while the pitch system is located between the blades and nacelle for adjusting the blade angle [24,25,26].

### 2.2. Composition and Classification of Wind Turbine Bearings

Bearings in wind turbines are used to provide physical support in the drive train and regulation system of the unit to decrease the friction and wear between components [10], which mainly include drive train bearings (main shaft, gearbox, and generator bearings) and regulating system bearings (yaw and pitch bearings) [14]. Figure 3 exhibits the installation position of bearings in the wind turbine.

The main structure and composition of each bearing are as follows:Main shaft bearing is the central rotating component of the wind turbine [28], which mainly supports the weight of the hub and blades. It requires transmitting the torque generated by the blade to the gearbox. Since the spindle bearing has to bear the alternating load generated by the blade torque and the gravitational load of the blade itself [29], the requirements for its machining accuracy, mechanical performance, and service life are extremely high, and the spindle bearing is required to have good alignment performance, smoothness, and vibration resistance [30]. Therefore, spindle bearings are usually selected from roller bearings with high load carrying capacity (for example: cylinder roller, spherical roller, or conical roller bearings) [31].Generator bearing is mainly designed to carry the high-speed rotation of the generator shaft. Cylindrical and spherical rolling ring bearings are commonly used as generator bearings. Through the structural design of these two types of bearings, the requirements of generators for high precision and low noise can be achieved [32,33]. Nevertheless, when selecting raw materials for the bearings, their insulating properties should be considered to avoid damage to the bearings caused by current pitting [34].Pitch bearing is being used to adjust the rotation angle of the blades for the purpose of regulating the windward direction of the blades and optimizing the electrical power delivery [35]. Failure of the pitch bearing can lead to loss of blade control, which can further affect the operational performance of the entire system of the wind turbine [36]. Due to the rotation of the hub and the instability of the wind direction and wind speed, the moment that the pitch bearing is subjected to also has time-varying and nonlinear characteristics [37]. Thus, the bearing mostly uses a four-point contact ball rotary bearing; this bearing groove is specifically machined for better wear and impact resistance and can withstand larger axial loads and higher moment loads [38,39,40].Yaw bearing is mounted between the tower and nacelle, which not only holds the weight, but also needs to adjust the windward angle of the wind turbine in time according to wind rotation [41] to optimize the power input and generation efficiency [42,43]. Since the yaw bearing is subjected to the impact load of the external wind and the internal overturning moment, the force situation is complicated. Consequently, the types of yaw bearings are generally the same as pitch bearings, i.e., one-row or two-row 4-point contact spherical bearings [44], for decreasing the wear of rolling contact surfaces [45].Gearbox bearing is a speed-increasing structure of wind turbines [46], and bearings with specific performance demands are selected for different positions. Thus, its bearings are comparatively large in number and contain various roller bearings and contact ball bearings, etc. [47]. According to the different load types, spindle support methods, and different gear stages, such bearings are provided to withstand shafts with different gear grades in high, medium, and low-speed stages [48]. For the low-speed stage, the bearing has to withstand the load from the main shaft, and cylindrical or tapered roller bearings with high load carrying capacity are typically used. However, medium and high-speed stage bearings only require the endurance of low axial and radial loads, so they regularly select 4-point contact ball, tapered roller, tubular roller, and others with lower loads capacity of bearings [49,50,51].Table 2 summarizes the specific application of bearings in different parts of wind turbines.

## 3. Fault Analysis and Research on Tribological Faults of Wind Turbine Bearings

The designed service life of wind power equipment is about 20 to 25 years [53], but due to factors such as unstable environmental conditions and time-varying loads, it cannot meet the service life standard [54]. Bearings as the central parts, their accuracy, performance, and lifetime have a decisive role in the performance and reliability of wind turbine facilities [55]. However, windmill machines perform under extremely unstable wind speeds, loading, and challenging operation situations, thus resulting in bearings being susceptible to damage during service [56]. It is estimated that only 10% of bearings operate properly during their life cycle, while 90% of bearing failures are due to problems such as poor lubrication (30%), improper maintenance (40%) and manufacturing defects (20%) failing to operate reliably during their life cycle [57,58]. The problem of bearing failure in wind turbines is one of the key reasons for the reliability life of wind turbines [59], which not only leads to loss of power generation but also increases the cost of repair and replacement parts [60].

For these reasons, it is essential to explore and study bearings to understand their failure features and patterns, which can effectively reduce maintenance costs [61,62]. In this paper, the common failure characteristics of wind turbine bearings will be explained in terms of common friction and lubrication problems.

### 3.1. Analysis of Common Frictional Failure Problems in Wind Turbine Bearings

Based on scholars’ summaries about the causes of bearing problems, various frictional failure modes may occur on wind turbine bearings, as shown in Figure 4.

Bearing frictional fault is caused by multiple failure modes together [68]. The following literature and ISO 15243:2004 [69] will be merged to classify and evaluate the failure modes of wind turbine bearings (See Figure 5 and Table 3).

Bearing failures can occur in one or more forms, initially due to surface damage caused by the fatigue of the material itself, but in severe cases it can lead to pitting corrosion and fatigue fractures on the bearing surface [73,74]. Normally the failure stages can be divided into three phases, which are crack formation, extension, and fracture [75]. As an example, Figure 6 was used to explain the bearing wear and friction mechanism using scanning electron microscopy and energy spectrometry to investigate the wear of the inner ring of the wind turbine bearing. From Figure 6a, it can be seen that the sides of the crater are perpendicular to the contact surface and have an irregular shape. A large number of cracks can be observed in and around the contact surface, which can be classified as surface rolling contact fatigue failure mode. The main chemical composition of the craters is consistent with the bearing material itself, with the presence of iron and chromium. The cracks formed in the initial stage do not affect the normal operation of the bearing, but if not maintained or replaced could cause the bearing to enter a more dangerous stage of extensive failure, resulting in destructive damage to the bearing. When the external impact load is high or the bearing is overloaded, it enters the second stage of failure expansion (Figure 6b), black particles appear in the crater, and the crack shape inside the bearing is more obvious. The bearing material has changed due to the generation of metal abrasive chips, and at this point the particle composition in the crater is predominantly molybdenum and sulfur. The rolling elements and raceways of the bearing also generate a slight plastic deformation. The rolling bearing micro-motion wear zone, due to the presence of wear and fatigue, will lead to increasing resistance to operation, resulting in vibration and noise. As shown in Figure 6c, the outer periphery of the inner ring of the bearing turns reddish brown and a large number of particles are present on the surface. EDS analysis further indicates that the chemical composition of the particles is mainly Fe_2_O_3_. These particles will further act as stress concentration points and accelerate the development of cracks and pits, leading to the formation of large irregular pits. When the abrasive debris generated during wear is mixed with water or lubricant, it causes resistance during bearing operation, which can lead to fracture failure of the bearing in severe cases [76,77,78].

Figure 7 shows the percentage of frictional failure modes in wind turbine bearings. The dominant forms of failure in bearings are wear, followed by plastic deformation [78,79]. In addition, issues such as spalling, corrosion, and friction-induced fracture are also major causes [80,81]. Fatigue failure and frictional damage of bearings is unavoidable, so how to take some methods to lower the trouble is an urgent problem to be resolved.

### 3.2. Analysis of Common Lubrication Failure Problems in Wind Turbine Bearings

In order to keep bearings in sound and stable working condition, it is necessary to ensure that the lubricant is in a reliable and effective working status in the first place and avoid the effect of its failure on the bearings [82]. The selection of each bearing of the wind turbine and its lubricant is shown in Table 4.

The function of lubricant is to avoid the direct contact between the rolling body and raceway [85], to form an oil film on the surface of the frictional subsurface, and decline the frictional heating inside the bearing. It minimizes the consumption of bearings when the equipment is under operation and stabilizes the structure to keep the wind power equipment running steadily [86,87,88].

In addition to wind power bearing failures caused by equipment deterioration problems, improper manufacturing, and material defects, about 70% of the failures are due to bearing wear and corrosion caused by lubrication failures [89,90]. Various lubrication failure modes may occur on the bearings presented in Figure 8.

Common causes of bearing lubrication failures in wind turbines include the following.

Physical factors of lubrication failure

The causes of lubrication failure in wind power bearings are mainly related to the influence of factors such as bearing speed, temperature, and load [94,95]. As the friction parts operate under high shear conditions, this makes the lubricant constantly subject to shear stress and centrifugal force, and causes destruction of the lubricant structure [96,97]. Thinning of the formed lubricant film will cause lubrication failure and lead to phenomena such as precipitation loss of lubricant out of the friction interface. If the temperature is overly elevated, the viscosity of the lubricant becomes low, and adhesion gets deteriorated. During running of the bearing, it is presumably thrown out of the bearing due to the centrifugal force, at which point the lubricant loses its effectiveness [64,98]. The process of lubrication failure due to physical factors is shown in Figure 9.

Chemical factors of lubrication failure

Since lubricants are readily oxidized by chemical reactions with oxygen in the air, the resulting organic acids corrode metal parts and lead to accelerated evaporation and black sludge generation, resulting in lower dropping points and poor visibility [100]. In addition, when wear particles (for example, copper, iron, nickel, and other bearing flaking particles) are mixed in the lubricant, these wear particles act as catalysts for lubricant oxidation and accelerate lubricant failure [101,102]. The process of lubrication failure due to chemical factors is demonstrated in Figure 10.

Contaminant factors of lubrication failure

Lubricant contamination is another major cause of wind bearing failure in turbines [104]. The most common sources of contamination for lubricants include abrasive chips, moisture, and dust [102,105]. When external contaminants enter the lubricant, they can cause damage to the internal fiber structure through physical and chemical effects, causing a decrease in the lubricant’s lubricating ability and resulting in a lubricant that does not achieve the desired lubrication effect [106]. Figure 11 shows the process of lubrication failure caused by contaminants.

Due to the seasonal temperature variations in the working environment of wind turbines, it is difficult for ordinary lubricants to maintain their long-term operation [107]. Consequently, the selection of lubricants should involve good high temperature adhesion and low temperature start-up, as well as high requirements for the viscosity index, cleanliness, and anti-wear properties of the lubricants [108,109,110]. Figure 12 presents a detailed description and summation for the causes of lubrication failures in wind turbine bearings.

From the above, after classifying and summarizing various failure types of wind turbine bearings, it is easy to notice that the main issues of bearing failure have changed from surface matters to lubrication problems [111]. According to statistics, most failures in wind turbines are related to bearings, and about 70% of bearing failures are linked to lubrication [112,113]. Regarding bearing faults, if researchers get hold of these failure modes and combine them with some effective lubrication techniques, O&M costs can be decreased, and power generation becomes more cost-effective [114].

## 4. Research on Bearings Lubrication Technology of Wind Turbine

In the last section, the problems about friction and lubrication failures of wind turbine bearings were discussed and analyzed. For decreasing bearing failure problems, it is highly meaningful to improve lubrication technologies for assuring stable operation and longer service life of wind turbine equipment [115,116,117]. Better lubrication not only protects the rolling element and the bearing itself by reducing friction, but also avoids corrosion and damage caused by the external environment [118,119]. Current technology allows for early detection of the initial stages of bearing spalling, but most bearings fail prematurely due to improper lubrication (contaminants, inadequate lubrication and lubricant selection) [120]. Moreover, 90% of bearings fail to attain their expected life due to lubrication problems [121]. So it is highly important that appropriate lubrication is applied to the bearings. This chapter summarizes recent research on the bearing lubrication technology of wind turbines, specifically on bearing lubricants, the method of bearing lubrication, and how to monitor the lubrication.

### 4.1. Research of Wind Turbine Bearings Lubrication Technology Based on Lubricants

Wind turbines are not ordinary mechanical equipment and have stringent requirements for lubrication [10]. Lubricants, as the “blood” of wind turbines, are considered to reflect the operating condition and life-span [122]. It can be said that the life cycle of wind turbines is closely related to the performance of lubricants [123,124,125]. Lubricants with superior performance can not only lower mechanical friction coefficients and bearing failures, but also have great influence on enhancing power generation efficiency in wind turbines [126,127].

So far, many scholars have studied lubricants for wind turbine bearings. Some researchers have studied a single grease applicable to bearings for minimizing mixing of lubricants in different parts of the wind turbine [128]. Sun [129] et al. synthesized a heterocyclic derivative additive with excellent sulfur content (RHY317) and compared it with the anti-wear additives already in use (RHY313, S-1, S-2) through four-ball testing machine and copper corrosion test. The results demonstrated that the extreme pressure of the additives’ anti-wear and anti-corrosion properties are better than the market products and can satisfy the performance requirements of wind power grease. At present, the extreme pressure anti-wear additives in wind power lubricants are mostly based on sulfur and phosphorus agents, with friction modifiers to improve the anti-micro-pitting ability of grease to meet the bearing lubrication requirements. Xia et al. [130] prepared a new type of wind turbine bearing grease (base oil is poly-olefin, thickener is lithium composite soap) by adding dibutyldithio-carbamate (T351) and thio-amino carbamate (T323) and other additives. The grease procedure is given in Figure 13. Through experiments, it is found that the tribological performance of the grease is better than the existing grease in the market, but the performance of the grease in practical application needs further study. Mutyala et al. [131] presented the concept of solid lubrication to reduce the periodic replacement and contamination of lubricants in bearings. They demonstrated that the formation of super-lubricated wear layers of carbon reduced overall friction for at least 20 times by using solid lubricant and two-dimensional MoS_2_ combined with Diamond-like-Carbon (DLC) films on a micro-pitting test rig. Compared to the oil-lubricated and steel–steel contact, there is no surface damage, reaching ultra-lubricity (traction friction coefficient of 0.003). This is further evidence that solid lubricants still have a lot of room for development. In order to decrease lubricant contamination and toxicity, Ng et al. [132] synthesized high oleic acid POME with graphene nanoparticles (GNP), multi-walled carbon nanotubes (MWCNT), and nano-structured graphite (NSG), respectively. Experiments showed that the graphene nanoparticles in high oleic acid had a significant reduction in friction and wear. This new base-oil with NSG provides a further decrease in fossil consumption. However, because of the high cost of nanoparticles, it requires further improvement of practical applications.

In the field of research on wind power main shaft lubricating grease, SKF Co. [133] developed a LGWM series about grease, which is based on low consistency mineral oil as an additive. Not only is it suitable for the requirements of wide temperature and extreme pressure of bearings, but it also has an excellent resistance to water and corruption. Gao et al. [134] used synthetic hydrocarbons as base oil, compound lithium soap as thickener, and supplemented with sulfur, phosphorus, and molybdenum agents, with amine and phenolic antioxidants blended. The rust inhibitor sulfonate and carboxylate were selected to create the grease suitable for the main shaft bearing of the wind machine. The developed grease was assessed by micro-wear, bearing life, bearing wear, and rust resistance tests. Experiment findings indicate that the lubricating grease has excellent wear and corrosion resistance.

As the pitch and yaw bearings have to withstand higher tilting torque and vibration, the lubricants are also required to be strictly demanding, which requires superior anti-abrasive characteristics and high extreme pressure performance [135]. According to the above, Gao et al. [136] developed a new type of wind turbine pitch bearing grease. The grease was made by thickening the base oil with 12-hydroxystearic acid/sebacic acid lithium soap compound and adding different proportions of antioxidants, metal deactivators, rust inhibitors and anti-micro-wear additives. The anti-wear performance, life-time and physicochemical properties of the developed pitch bearing grease were demonstrated to reach the application level through experiments. The subsequent practical application also met the excellent lubrication demands of the grease, which also providing new ideas for the future wind power grease development. Schwack et al. [137] tested six industrial oils with different compositions in order to study the anti-wear performance of fan variable pitch bearing grease. The results indicate that the oil lubricant with low base oil viscosity and high seepage rate has the best wear resistance. Nonetheless, the six lubricants all cause wear under experimental conditions, so further research is required in the development of grease.

In the field of wind turbine gear oil research, Sun et al. [138] tested two gear oils containing phosphorus additives (L1-320 and L2-320). The amount of variation of iron elements in wind turbine gear oils was evaluated by spectral analysis technique. After statistical analysis, it was concluded that L1-320 had better extreme pressure-anti-wear properties than L2-320. Saidi et al. [139] prepared MoS2 nano-lubricant additives. The friction coefficient of wind turbine gear oil added with MoS2 is shown in Figure 14. It is proved that the additive has high wear resistance and anti-friction ability by tribometer and wear test. Iglesias et al. [140] used ionic liquids (IL), which are considered “green” solvents, as lubricant additives to decrease the wear behavior of wind turbine gearbox bearings and gears. The research was conducted by adding different concentrations and ratios of IL with synthetic poly α-olefin to create lubrication compounds. The tribological experiment was performed on a tribometer with an AISI 52,100 steel flat plate and AISI 440C steel ball. Experiments have proven that adding 5–10 wt.% of IL to the base oil will reduce friction, which also implies that IL, as a non-toxic and non-polluting gear oil lubrication additive, can provide better anti-wear effect compared with existing gearbox lubricants. The optical micrographs of the wear surface after the addition of ionic additives are given in Figure 14. However, further research is required for the use of ionic liquid as a pure lubricant for wind turbine lubrication.

In the development of lubricants for wind turbine, the performance of lubricants such as oxygen resistance, water resistance, mechanical stability, and rust resistance are the priority considered characteristics. Although the current research on lubricants has made great progress, it is still insufficient. With the large-scale construction of offshore wind farms, new requirements are put forward for the lubrication of wind turbines, so the lubricating grease for offshore wind farms should be the focus of attention of scientific and technical personnel in recent years.

Concerning the application of self-lubrication of bearing lubricants, Ali et al. [141] incorporated TiO2 and TiO2/G hybrid NMs in M50 high-temperature bearing steel matrix. The results showed that frictional wear experiments at different temperatures demonstrated the formation of self-lubricating films in the friction chemistry during the friction process, which further suggested that solid lubricants have excellent friction-reducing properties at high temperatures. Elsheikh et al. [142] combined two nano self-lubricating materials (SnS and/or ZnO) with M50 matrix in the ratio of 10% to make two new matrix composites M50 + 10% SnS(MS) and M50 + 10% SnS + 10% ZnO(MSZ). The tribological properties of the two samples were investigated at different loads, velocities, and temperatures, and it was experimentally demonstrated that the friction of MSZ was the smallest at high temperatures, which also proved that combining different solid nanolubricants with each other would have a positive effect on improving the tribological properties of the bearing steel. Figure 15 showed that for high temperature, MSZ is more advantageous. Essa et al. [143] studied the tribological behavior of composites MZ (M50 + ZnO), MM (M50 + MoS2), and MZM (M50 + ZnO + MoS2). It was discovered that MM had superior lubricating properties below 400 °C and MZ had better lubricating characteristics above 400 °C. Because of the synergistic impact, MZM possesses the best tribological properties over a wide temperature range. This demonstrated that using a high proportion of mixed solid lubricants reduces friction coefficients and increases bearing service life. After that, Ali et al. [144] re-investigated the application of different dimensions of Al_2_O_3_ and TiO_2_ nanoparticles as lubricant additives to reduce tribological behavior. The results showed that the friction coefficient, power loss, and wear rate were reduced when 0.25.% concentration of nanoparticles was added to oleic acid compared to the lubricant without nanoparticles.

When investigating the effect of additives in lubricants, Li et al. [145] studied the relationship between grease and bearing surface materials by selecting the grease of wind turbine bearing parts. In addition, frictional wear experiments were conducted under different loads. This research verified that extreme pressure and solid additives in grease could contribute to corrosion of bearing steel surfaces. To examine the effect of metallic and non-metallic additives in lubricants on the lubrication effect, Haque et al. [146] conducted experiments with the thrust bearing test bench. The findings indicated that lubricants with metal additives have inferior tribological properties and form inadequate lubricant films, which are highly prone to water contamination and lead to white etched cracks (WECs) during bearing operation. Therefore, it also proves that when lubricating different parts of wind power bearings, grease with different additives should be selected to lubricate them.

About the replacement time of wind turbine bearing lubricant, Li et al. [147] monitored and investigated the indicators such as drop point, ferromagnetic particle density, mechanical impurities, and metallic elements of grease from multiple wind turbines. It is proposed that the lubricant should be exchanged for bearings with dropping point falling more than 30 °C and iron content over 5000 ppm. On how to dispose of waste lubricants, Feng et al. [148] heated wind turbine gearbox lubricants at 60 °C and used polarity-inducing adsorbent for adsorption of impurities as the recovery treatment. Subsequently, the lubricants reached the same performance as fresh oil, which provided a new way of dealing with old-fashioned lubricants.

The influence of lubricants on bearing performance is self-evident [87,149]. Lubricants not only lower wear and friction of bearings, but also provide a protective layer against dust and corrosion that acts as a heat sink at the bearing operation [150]. For onshore wind turbines, extreme pressure and wear resistance of lubricants should be considered in lubrication, while in offshore windmills, corrosion resistance, and wide temperature, resistance of lubricants should be principally considered [109,151]. For different bearing speeds, the requirements for lubricant dosage are also different [152]. With high and mid-speed bearings, the choice of grease lubrication should be about 20–30%; in contrast, for low-speed bearings, it ought to be roughly 80%. When selecting oil lubrication, it is fitting to employ low-viscosity lubricant for fast-speed bearings, while the low-speed with high load or high-working temperature is more proper to select high-viscosity lubricant [153,154,155]. Appropriate and properly applied lubricants can not only reduce maintenance and lubrication costs, but also reduce wind turbine downtime and improve the reliability of equipment [156].

Table 5 describes a summary of researchers’ research on lubricant-based lubrication technology for wind turbine bearings.

### 4.2. Research of Wind Turbine Bearings Lubrication Technology Based on Lubrication Methods

The smooth operation of wind power bearings cannot be achieved without the lubrication of lubricants [157], but the lubrication method of the bearings also has a major impact on the life-cycle. The majority of current lubrication methods are timed and quantified, which is highly likely to result in under- or over-lubrication of bearings, thereby leading to wear or non-ferrous metal reactions and lubricant spillage contamination [158,159]. Appropriate lubrication methods can make the bearing lubrication more effective [160] and further extend their service life. The current common methods of lubrication for bearings and its advantages and disadvantages are shown in Table 6.

The most common lubrication method for wind turbine bearings is still the centralized lubrication system [163]. However, there are still many problems in the application of centralized lubrication system on wind turbine bearings, for which many scholars have explored other different lubrication methods.

Yao et al. [164] developed a new centralized lubrication pumping station. The device improves the fluidity of grease by adding oil absorption structures such as thin oil vanes and oil delivery impellers, connecting the evolved gravity balance oil level sensor to signal module for monitoring the viscosity of grease. This pump station has gained better results in realistic wind power lubrication system. It has reduced the dependence of centralized lubrication devices on viscosity and flow of grease and solved the problem of grease deposition and deterioration in extreme cold environments. However, when this pump station is agitating the high-viscosity lubricants, the structure of the grease is susceptible to destruction, which has a certain impact on lubricants’ performance. Wang et al. [165] built a wind turbine bearing lubrication condition model with elastic flow lubrication theory based on Hertz contact [166]. Through this model, and by contact analysis on the roller-race of bearings, the lubrication state of the bearing can be precisely confirmed. However, the model has only been simulated and needs to be further explored for ap-plication in practice. Gu [167] redesigned the structure of rolling bearings. The upper and lower part of the slider of the rolling bearing is designed as an arc with a certain curvature (as shown in Figure 16). Numerical calculations and FEM analysis were applied to research the lubrication characteristics of the contact surface between the slider and the raceway. Modeling results show that:The oil and gas mixed lubrication mode forms an effective lubricant film in the contact area.Lubricant accumulation is caused by the oil and gas pipe clamping angle and excessive air inlet velocity and small oil inlet velocity.The adhesion effect of oil droplets on the slider is determined by the lubricant viscosity.

However, the design of this slider structure does not take into account the effects of pressure, film thickness and temperature, and the oil–air two-phase flow lubrication test needs to be further explored.

Peng et al. [168] added an automatic temperature-controlled heating device in the grease distribution design in a bid to adjust the wind power bearing lubrication system dispenser low-temperature prone to blockage. This system mainly provides temperature control in the lubrication system, heating the main distribution valve and controlling the grease at normal working temperature in the valve. Afterwards, the heated grease is delivered by a sub-distribution valve to each lubrication point of the bearing. Nevertheless, the temperature-controlled heating device in the installation of the wiring is more complex, need to consider certain cost issues. Meng et al. [169] created a discrete element numerical model for particle flow lubrication based on a novel particle flow lubrication mode (shown in Figure 17). It was further indicated that the fluctuation velocity of the particle-lubricated medium is a key parameter that directly reflects the fast or low macroscopic flow rate.

Regarding the use of wind turbine bearing micro-nozzles, Westerberg et al. [170] used particle image velocimetry to estimate the grease flow in the elbow channel when using wind turbine bearing micro-nozzles. Grease flow in the elbow channel was investigated based on bearing grease rheology and flow conditions, with the effect of pressure on the flow in the channel simulated by establishing a rheological model. The results of the experiment revealed that the elbow angle of the nozzle is related to the grease flow rate and a high flow rate reduces the sliding impact. Following that, Liu et al. [171] created a fluid–solid coupled simulation model to explore the impacts of oil flow rate, lubricant viscosity, nozzle angle, and number of nozzles on gearbox bearing lubrication characteristics. The results of using the CFD method to simulate the air–oil two-phase flow in the bearing revealed that the oil flow rate has little effect on the minimum oil volume fraction of the bearing, that increasing the nozzle angle and lubricant viscosity decreases the lubricant passage rate, and that the lubrication effect is better in the case of double nozzles. This has some implications for enhancing the lubricating properties of gearboxes. Li et al. [172] designed a bearing lubrication device based on the piezoelectric microjet theory, and its finite element model is shown in Figure 18. The device utilizes the limited space of the bearing system to achieve on-demand active lubrication by monitoring the bearing temperature and friction torque. Moreover, it can reach high-efficiency lubrication. Farré-Lladós et al. [173] devised and built micro-nozzles (shown in Figure 19) to automatically lubricate wind turbine pitch gears. The wear coefficient between the spinning elements was lowered by 70% by evaluating the pitch system on the test bench. This invention has greatly aided future research into self-lubricating bearings. However, it is still difficult to manufacture micro-nozzles. Oliveira et al. [174] introduced a bearing-based rotating system model for modeling and identifying the oil supply in a dynamic pressure bearing. Based on this model, a freshly identifiable technology for bearing lubrication was developed. The technique employs the rotor vibration signal of bearings and the oil supply flow rate as input parameters of lubrication model for analysis of bearing lubrication quantity. Numerical simulations show that the method can effectively diagnose the amount of oil in a bearing under both under-lubricated and oil-soaked conditions.

Ge et al. [175] proposed a groove structure (semi-elliptical with a long semi-axial length of 0.5 mm) to the non-contact region of the bearing inside race surface to increase the efficiency of jet lubrication on bearings. This axial structure of the groove was designed in the zone between the nozzle and the roller surface of the inner ring. Through the oil flow lubrication analysis and lubrication performance experiment of the improved bearing, it is observed that the groove structure added in the inner ring of the bearing can guide more lubricant into the lubrication area of the bearing raceway. In addition, bearings with groove structure also have a lower temperature increase, which can effectively improve the lubrication performance. The axial groove structure and its position in the bearing design is shown in Figure 20.

Erill et al. [176] invented a dynamic lubrication system for wind turbine pitch bearings. It is designed to lubricate the pitch bearing when it needs to be greased according to the operating conditions, thus avoiding damage to the bearing caused by excessive or insufficient lubrication. The system is dependent on a programmable control unit to judge the rotational conditions of the pitch bearing so that it can determine the lubrication period and interval. Yang et al. [177] proposed dynamic lubrication control for spindle bearings. The output power of wind turbine is taken as the calculation principle for dynamic lubrication, real-time operation data, and bearing load are regarded as the parameters change of lubricant supply, and wind power, friction of bearings, in addition to the revolving speed, are used to determine the lubrication parameters. A time sequence-based self-adaptive algorithm of smoothing prediction is used to calculate the lubrication demand of main bearings, and then dynamic lubrication of wind turbine main shaft bearings is carried out. This dynamic lubrication control strategy can be expressed as:(1)$$\left\{\begin{array}{l} M_{t}=\frac{{\hat{y}}_{t}}{P_{r}}\times M,{\hat{y}}_{t}\geq P_{a}\\ M_{t}{}_{1}=0,M_{t2}=M,\;{\hat{y}}_{t}{}_{1}<P_{a},\int_{t1}^{t2}{\hat{y}}_{t}dt=P_{a} \end{array}\right.$$

In case bearing requires lubrication, $P_{a}$ is the critical value of wind power, M(ml) is quantity of lubricants added at the $P_{a}$ weight. By analyzing the historical data of wind turbine output for two days at the beginning and end of the month at a wind power plant for both timed and non-quantitative lubrication, it has been demonstrated that the method could achieve flexible lubrication control of wind turbine.

In choosing a suitable method of lubrication for wind turbine bearings, first consider the possible effects of external environment and internal friction and other factors on lubrication, followed by the consideration of the internal structure of the wind turbine equipment and transmission process [178]. It is advisable to combine advanced lubrication materials to further select a more suitable lubrication method, and to achieve green and effective lubrication of wind turbine bearings [179].

Table 7 shows the summary of researchers’ research on wind turbine bearing lubrication methods.

### 4.3. Research of Wind Turbine Bearings Lubrication Technology Based on Lubrication Monitoring Methods

Although there has been considerable research on the bearing aspects of wind power generation, there is still a threat of premature bearing failure problems, most of which are also related to lubrication [180,181]. Furthermore, the choice of lubricants, lubrication intervals, the number of lubricants, and lubrication decontamination are also essential factors that impact upon the failure of lubrication [182,183]. Subsequently, if real-time lubrication status monitoring and fault identification can be executed on bearings, the amount of lubrication and lubrication interval can be regulated more accurately, thus preventing lubricant contamination and making lubrication more efficient.

The lubrication state of wind power equipment is generally reflected by the concentration of wear particles in the lubricant [184]. In the beginning phase of failure, poor lubrication can lead to particle wear or spalling of the bearing [185], when the characteristics of the failure can be identified using acoustic signals to distinguish whether the lubrication of the bearing is abnormal. Acoustic detection methods can accurately identify the concentration of solid particles in lubricants [186]. The current acoustic detection method used for bearing lubrication monitoring generally uses acoustic sensors to receive signal echoes of irregular vibrations generated by lubricant abrasive particles and determine whether the bearing lubrication has failed based on the amplitude of the signal [187,188]. When analyzing the lubrication status of bearings, Su et al. [184] combined the monitoring of bearing lubrication status with fault diagnosis, analyzing the ultrasonic signal characteristics of the bearing, and responded the lubrication status of the bearing with the signal amplitude diagram (Figure 21), which was compared and evaluated with the bearing fault degree. It was experimented that the effective value and sort entropy of ultrasonic signal display can be used as characteristic quantities to reflect the bearing faults, and the lubrication and operation faults of bearings can be directly expressed by the waveform and histogram of ultrasonic signal.

Similarly, Mijares et al. [189] monitored the amount of lubrication inside the bearing by a method related to vibration analysis. The method tested the bearing for preload and minimum lubrication level and performed a comparative analysis of indicator performance for kurtosis (pulse in the signal), CF (crest factor), and RMS (root mean square value of the vibration signal). It was proved through experiments that the increase of preload can reduce the signal pulse, and the change of kurtosis derived from vibration analysis can detect the adequacy of lubricant inside the bearing. The test rig used to conduct the experiments is shown in Figure 22. The experiment used a syringe to add lubricant to the bearing and samples the vibration signal by loading the bearing at a certain frequency. A torque meter with a speed sensor monitors the torque signal generated by the rotation of the bearing. By testing different levels of lubricant to evaluate the characteristics of the bearing vibration signal under dry, low, and fully lubricated conditions, the acquired signals are processed by digital filters to eliminate interference from external factors. It has been proved that the kurtosis ratio CF can distinguish more clearly between lubrication and drying conditions inside the bearing.

Nicholas et al. [190] modified the ultrasonic reflection method by proposing a new ultrasonic reflection method to monitor the variation of oil volume amongst bearing rollers in wind turbine gearboxes. An ultrasonic sensor mounted on the outside of the bearing raceway (see in Figure 23) was used to measure the minute vibrations generated by the ball of the bearing raceway to obtain the reflection coefficient R, which is used to determine the lubrication status of the bearing roller raceway. Through the data collected in test benches and field, various lubrication conditions such as under-lubrication, part-lubrication, or fully-lubrication were accurately monitored during the operation of the bearings. Nevertheless, most of the measurements were measured during transient operation and are not up to standard for steady-state operation. Ibrahim et al. [191] conducted experimental tests of grease in bearings with different concentrations and particle sizes of solid contaminants by using acoustic emission technique. Compared with the characteristic signals of bearings in normal operation, it is demonstrated through experiment that the root-mean-square and kurtosis values of the AE wave energy characteristics of the acoustic emission signal increase when there are larger particles in the lubricant. The size and concentration of contaminants in the lubricant could be studied in detail by the RMS and kurtosis values of energy levels in acoustic emission signals.

The most common methodology for health monitoring of bearing lubrication is still based on the analysis of bearing vibrations on acoustic signals [192]. The frequency, signal characteristics of bearing vibrations, and the decibel levels generated during operation vary when there are abnormalities in lubrication [193]. However, since the decibels generated during bearing operation can be shown by acoustic devices, the decibel level will decrease as lubricant is added to the bearing, and the waveform of the acoustic signal generated by the bearing vibration will tend to plateau at this time [194]. However, the acoustic detection method still monitors the lubrication status with the help of bearing diagnosis, and its accuracy and timeliness are not accurate enough [195,196,197]. Therefore, other lubrication monitoring sensors and methods are also summarized in this paper.

In the area of research on machine algorithms for lubrication condition monitoring, Ünlü et al. [198] used an artificial neural network algorithm and the EBP technique to determine the frictional wear behavior under bearing lubrication conditions. For monitoring the lubrication condition of bearings, the machine learning approach used a 3:4:4:3 multilayer structure, with time, load, and speed as input vectors and friction coefficient, journal, and bearing weight loss as output values, which were later input into the dataset for calculation. The experimental comparison on the bearing test stand revealed that the method’s forecast findings matched the experimental data. The trained ANN values, according to the authors, can efficiently determine the friction coefficient and wear loss of the bearings. Moder et al. [199] showed that machine learning (ML) algorithms can be used to predict the lubrication status of high-speed torque bearings. This lubrication status classification method (shown in Figure 24) collected the high-speed torque signal of the bearing by using the fast Fourier transform (FFT) and used it to analyze the time series of the torque. The time series of the signal was used to accurately predict the lubrication status of the bearing using two ML algorithms (artificial neural network and logistic regression). The results of this survey provided a basis for further research on machine learning algorithms that can be applied to the measurement of bearing sensors. Zavos et al. [200] developed an ML model based on an ordered regression algorithm in order to predict the coating application and lubricant selection for bearing components. A tribological analysis model was developed with various lubricants and coatings for bearings as parameters and the models with different regression methods accurately predicted viscous and boundary friction of the bearings. The article demonstrated that the use of a second-order polynomial regression model was more accurate in predicting the coefficient of friction and that the wear on the surface of the coating containing nanoparticles was less. Katsaros et al. [201] integrated numerical analysis and machine learning techniques. The Reynolds equation was solved using the finite difference method, the hydrodynamic lubrication characteristics of the bearing were used as the output, and a prediction model was developed to determine the most suitable lubricant for the bearing. The authors revealed that the multivariate model predicted with higher accuracy.

In the area of developing lubrication-based monitoring models, Márton et al. [202] proposed an energy balance method for viscosity detection of lubricants based on the relative relationship between friction parameter pairs and the lubricant. The proposed method for monitoring the health of lubricants has been implemented (see Figure 25). The approach consisted of a speed-controlled and current-controlled synchronous motors. By using the position and temperature sensors installed in the housing to determine the shell temperature and mechanical system speed, which further identifies the temperature of the lubricant. The variable temperature and viscosity of the lubricants were used as parameters to judge the residual life of lubricants based on the basic model of Coulomb + Viscous Friction, which was coupled with the health monitoring algorithm. In addition, by designing a fault detector and dynamic model of the lubricant, an in-depth study of lubricant life monitoring based on friction parameter estimation was carried out.

However, the current monitoring of lubricant viscosity based on the lubrication monitoring model is not accurate, and in the subsequent research, attention should be paid to the selection of suitable lubrication parameters to improve the model.

In the development of lubricant monitoring devices and sensors, many scholars have designed lubricant monitoring sensors and corresponding devices to monitor the oil for degradation, contamination, and corrosion. Murali et al. [203] designed a microfluidic device that enabled the statistical detection of metal particles in low conductivity lubricants based on the Coulter counting principle in order to detect wear particle debris in lubricants. Shown in Figure 26, the device consists of a reservoir (inlet and outlet), a single fluid channel (40 μm (h) × 100 μm (W) × 300 μm (L)), and coplanar electrodes (spacing of 20 μm (s)). While metallic wear particles in the lubricant flow through the microchannel, the capacitance between the electrodes changes due to differences in dielectric constants, and this microfluidic capacitance sensor can individually detect wear debris from 10 µm to 25 µm. However, the device did not distinguish between metals and other types of particles because the debris grains all have similar dielectric constants, and the microchannels made the monitor capability poor. To improve the failure of lubricants, Dittes et al. [204] made a corrosion sensor (Figure 27). The sensor consisted of a primary cell with chemically plated nickel-impregnated gold and a zinc electrode. When the lubricant becomes corrosive, the current between the metals measured at the zinc electrode changes. The current is a proportional value to the corrosion rate, which is used to forecast the corrosion rate of the contaminated lubricant and the water contamination in the lubricant. However, the sensor also had a certain bias on the measurement result because the plated metal would interact with the contaminants in the lubricant. Therefore, the sensor needs more in-depth study in practical applications.

Currently, a grease condition monitoring tool has been developed by Schaeffler Co. [205] (as indicated in Figure 28). The grease condition monitoring tool consists of a tiny grease sensor and an electronic evaluation system. A miniature sensor (5 mm diameter) is inserted into the bore of a rolling bearing and comes into contact with the lubricant. The sensor irradiates the grease with specific wavelengths in the infrared spectrum and measures the water content and temperature within the lubricant by means of an optical infrared reflection method. Finally, the signal measured by the sensor is transmitted via a cable to an evaluation unit (electronic evaluation system). However, since corrosion of lubricants increases with water content and oxidation [206], this detection tool is only a rough estimate of the water content in the lubricant by optical measurements and has limited usability. In addition, airborne debris can cause incorrect measurements, and the depth of the sensor can also affect the results.

For the variation of water contents in lubricants, sensors for monitoring the real-time moisture levels have also been widely developed [207,208,209]. Raadnui et al. [210] showed a cost-effective gate capacitance sensor to measure contamination of water and other impurities in lubricants by changes in the dielectric constant. Such a sensor consists of two magnetic poles connected in series, and its transducer head is used to measure the change of water amount. If the lifetime of lubricant is surpassed, its function will be impaired, so how to determine whether it deteriorated is still a thorny issue at present [207,211]. Karluk et al. [212] proposed a method for observing lubricants degradation based on frictional charging in order to instantly monitor the deterioration phenomenon. Figure 29 shows the polymer frictional electro-oxidation sensor for lubricant monitoring process. This method reflects the oxidation of the lubricant by the change in the charging signal resulting from the contact (separation) of the oil-impregnated cellulose and polymer phases, i.e., when the potential is reduced, the lubricant is oxidized. Maruyama et al. [213] developed an electrical impedance method for monitoring the lubrication status of deep groove ball bearings. The methodology was primarily for monitoring the oil film thickness and breakdown rate in elliptical elastomeric flow (EHD) contacts of bearings. The breakdown ratio and the oil film thickness of the bearing were quantitatively evaluated through applying a positive voltage to the contacts in the elastomeric contact from the resulting compound resistance. By monitoring actual ambient temperatures, the method was found to measure oil film depth in the low-velocity range, which is consistent with the theoretical values of the Hubrock–Dowson model. However, the shear heating and lubricant consumption at high rates cause the measure film to be thinner than the theoretical value. Details of the test procedure for this method are shown in Figure 30. The test was carried out by applying two bearings installed on a revolving shaft. The lubricant was initially injected into the bearings by applying a sinusoidal voltage (RMS amplitude: Ve = 1.5 V, frequency: f = 1.0 MHz) through an LCR meter between the rotating shaft and the bearing housing. Meanwhile, a spring was used to apply an axial load Fa, and the rotating shaft rotates while driving the rotational speed of the inner ring N, and at this point, the bearing outer ring temperature (thermocouple directly connected to the bearing outer ring) and torque (load cell measurement), as well as the average oil film thickness h and breakdown ratio α, can be measured. 

In the field of wind power, the research on lubrication monitoring sensors has become a hot topic at present. However, the issue of electromagnetic compatibility in sensors is still a challenging work. Due to the frequent phenomenon of high currents in wind power, changes in the external magnetic field can also affect the monitoring results of the sensors. Therefore, in the future work, attention should be paid to the combination of the corresponding measurement principles (such as photoelectricity, laser, and other advanced means) to enhance the anti-interference ability and structural stability of the lubrication sensor.

Table 8 shows a summary of the researchers’ research on wind turbine bearing lubrication monitoring methods.

Real-time monitoring of bearing lubrication is the most direct and effective method to determine the health of bearings, and although many explorations have been conducted in both academia and industry for surveillance techniques [214,215], further research is still required. Most of the existing research findings are still based on the signal analysis of bearing vibration so as to detect lubrication status in time and carry out maintenance when abnormal fluctuation of shock data occurs. Although it can prevent operative lubrication failure and other situations, the noise generated by the bearings during operation will also affect the acquisition and analysis of signals, and this will bring some difficulties to the detection. In future research, it is required to evaluate the correlation between monitoring methods of bearing lubrication and oil data along with lubrication approaches, combined with intelligent control and corresponding sensor monitoring technologies to achieve healthy bearing operation.

## 5. Conclusions and Future Perspectives

### 5.1. Conclusions

Along with the fast-growing wind power generation industry, the annual power generation of wind turbines is steadily increasing, and wind power is already the most promising energy source [216]. However, the cost of wind energy production is exacerbated by premature failures of its turbine bearings [9,217]. As mentioned above, to achieve cost-effectiveness of wind turbines is to prolong the service life of bearings as long as practicable [218,219]. There are three key principles to maximizing bearing life: selecting the correct lubricants, administering appropriate lubricants, and maintaining lubricants in a clean condition (corresponding to parts 1, 2, and 3 in Chapter 4 of the paper). Any negligence in these three areas will increase the risk of premature bearing failure and disrupt the normal operation of the equipment.

This paper first reviews the structure of wind turbines and the types of bearings and analyzes the common causes of friction and lubrication failure. On this basis, it collates the research progress in the lubrication technology of wind turbine bearings in recent years. Finally, a summary of existing problems and the outlook for future research directions is presented. In summary, experimental and theoretical research in the following areas is still needed in the future:Research on wind turbine bearings failure analysis

In recent years, researchers have studied the failure of wind power bearings and understood many failure modes and causes, but there are still many challenges in the study of tribological problems of bearings. For complex failure modes, it is a challenge to determine the root causes, and research on main shaft, pitch, and generator bearings is still insufficient. Therefore, more basic research on wind power bearings is still needed to fully understand the failure mechanisms and damage modalities of bearing failures.

Research on wind turbine bearings lubrication technology

The study of wind power bearing lubrication technology involves comprehensive consideration of various factors, including bearing operation, lubricant materials, the application of lubrication methods, and monitoring the lubrication condition. The speed of bearing will affect the flow distribution state of lubricant, the material of lubricant is the direct factor in whether the lubrication fails or not, the use of lubrication method is the premise to ensure the adequate lubrication of bearing, and the real-time lubrication monitoring of bearing is the guarantee to avoid the lubrication failure.

### 5.2. Future Perspective and Recommendations

Research on lubricants for wind power bearings:

When it comes to lubricants, nano-lubricating materials still have a lot of room for development. In the application of grease, the impact of additives on the overall lubrication should be thoroughly considered. With regard to the failure of the lubricants, appropriate lubrication diagnostic techniques can be evolved and combined with applicable bearing diagnostics to observe indicators such as moisture, particles, and temperature.

Choosing the right lubrication method:

To adequately consider the operating condition of the equipment and lubrication interval, avoiding excessive or insufficient lubrication, as well as a comprehensive inspection and monitoring of the lubrication status of the equipment, is similarly critical. Lubrication methods can be combined with big data and computer technology to more accurately analyze the data and lubrication status of bearings.

Bearing lubrication monitoring:

The most common method used is still the acoustic detection method, based on abnormal bearing vibration data to determine the lubrication status of bearing lubrication. However, the vibration signal is easily disrupted by other noise factors, which can be combined with the corresponding oil monitoring technology for more accurate monitoring of the lubrication status of the bearing. When developing the corresponding lubrication monitoring sensors, attention should be paid to the possible effects of water and contaminants in the lubricant, and it is better to improve the database of lubrication data in conjunction with the operating condition of the bearings and tribological data, thus making the measurements more accurate.

With the high growth of the wind turbine industry, future research on wind power bearings should be developed in the direction of digitalization and intelligence. For problems such as lubricant deterioration and contamination, relevant lubrication sensors can be developed to monitor indicators such as particles, moisture, and dust in the lubricant in a uniform manner, in addition to combining bearing vibration data with oil analysis data. New lubricant formulations and additives need to be developed to enhance the properties of wind turbine bearing lubricants. Regarding the selection of bearing lubrication method, it should be coupled with some intelligent control approaches to determine the wear status of bearings, and use the platform such as Big Data and Artificial Intelligence to analyze the lubrication status accurately so as to avoid the impact of excessive and deficient lubricants on the bearing. On this account, the stability of wind power generation equipment can be improved, and the green lubrication of wind power bearings will be achieved.

## Figures and Tables

**Figure 1 polymers-14-03041-f001:**
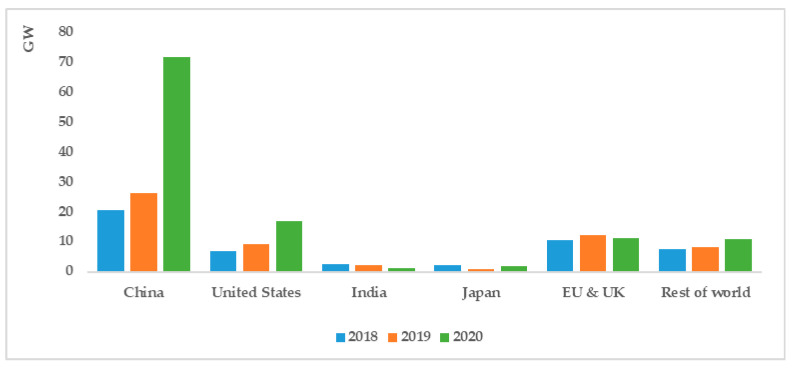
Global net annual increase in wind power generation 2018–2020 [5].

**Figure 2 polymers-14-03041-f002:**
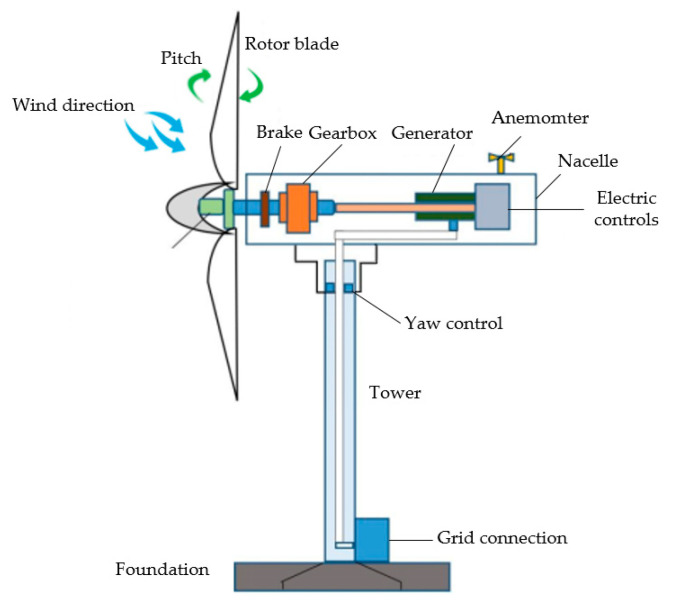
Three-blade horizontal axis wind turbine [23].

**Figure 3 polymers-14-03041-f003:**
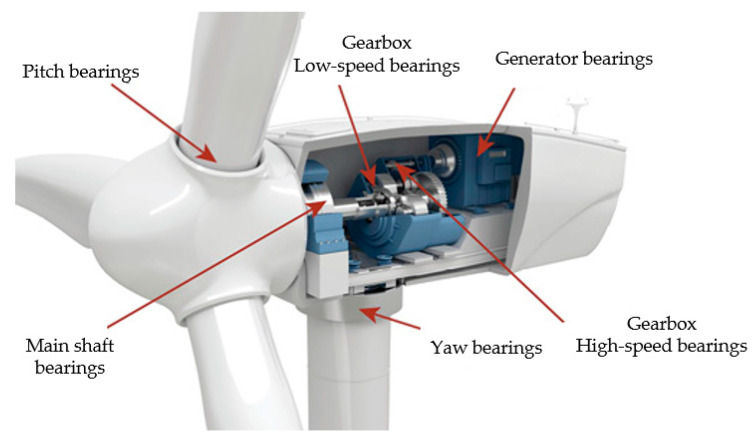
Bearings mounting position [27].

**Figure 4 polymers-14-03041-f004:**
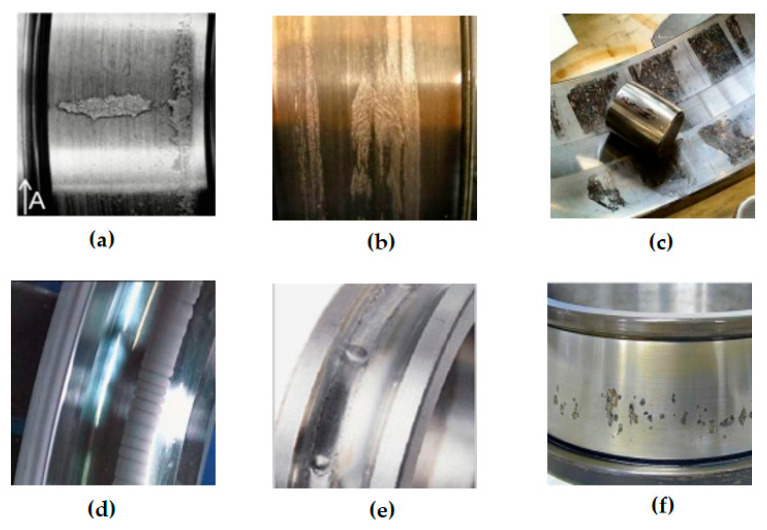
Forms of frictional failure of wind turbine bearings, (**a**) Bearing fatigue failure resulting in surface rolling contact fatigue; (**b**) Adhesive wear in bearing wear types; (**c**) Bearing corrosion resulting in moisture corrosion; (**d**) Bearing galvanic corrosion due to excessive current; (**e**) Plastic warpage of the bearing due to excessive load; (**f**) Bearing surface fracture cracks (Adapted with permission from Refs. [63,64,65,66,67]. Copyright 2016 Elsevier).

**Figure 5 polymers-14-03041-f005:**
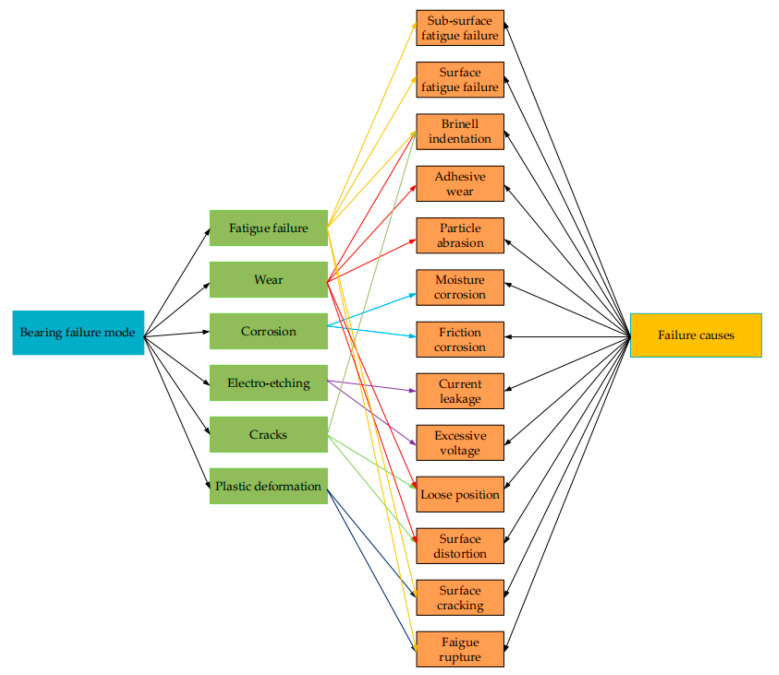
Classification of bearing friction failure modes in wind turbines (Adapted from [69,70]).

**Figure 6 polymers-14-03041-f006:**
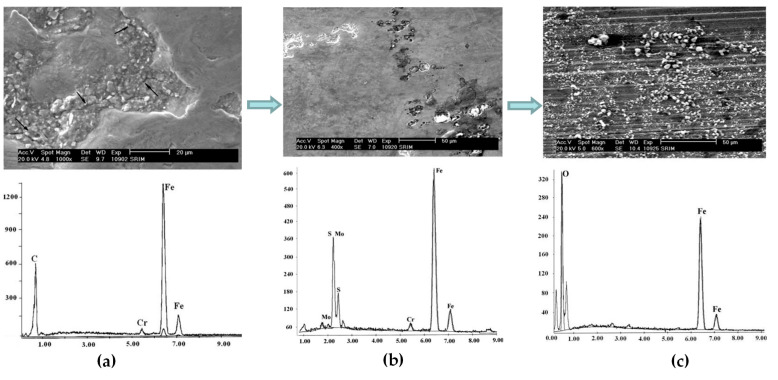
SEM and EDS of inner ring wear of bearings (Reprinted with permission from Ref. [76]. (Copyright 2011 Springer).

**Figure 7 polymers-14-03041-f007:**
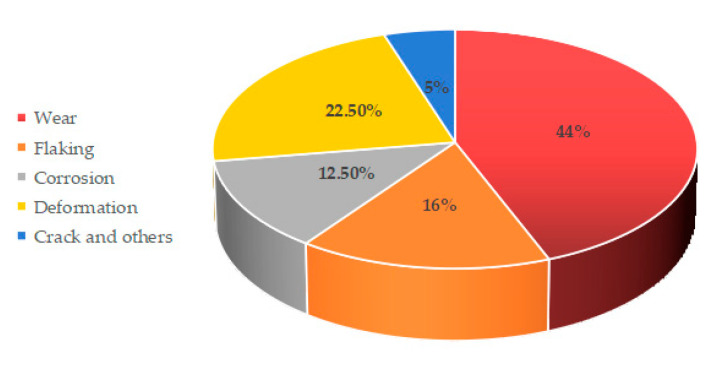
Percentage of frictional failure forms in wind power bearings (Adapted from [63]).

**Figure 8 polymers-14-03041-f008:**
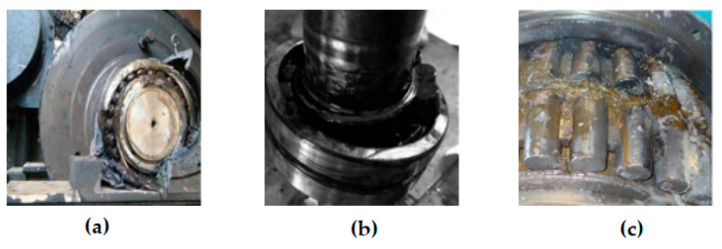
Bearing Lubrication Failure Mode [91,92,93]. (**a**) Bearing failure due to excessive lubrication; (**b**) Bearing failure due to lubricant deterioration; (**c**) Bearing failure due to insufficient lubrication.

**Figure 9 polymers-14-03041-f009:**
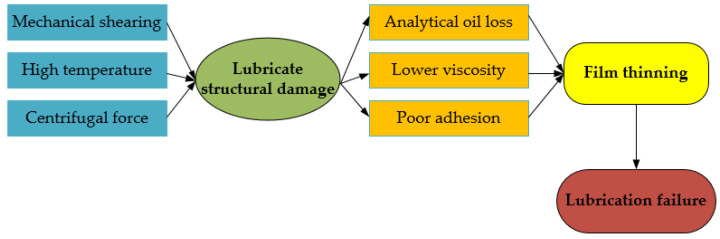
Physical factors of lubrication failure [98,99].

**Figure 10 polymers-14-03041-f010:**
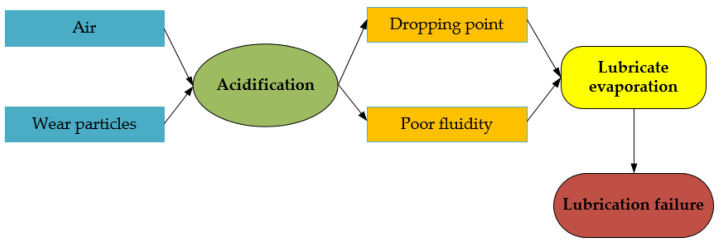
Chemical factors of lubrication failure [99,103].

**Figure 11 polymers-14-03041-f011:**
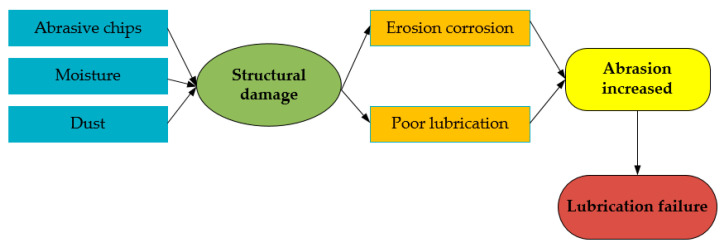
Contaminant factors of lubrication failure [104,106].

**Figure 12 polymers-14-03041-f012:**
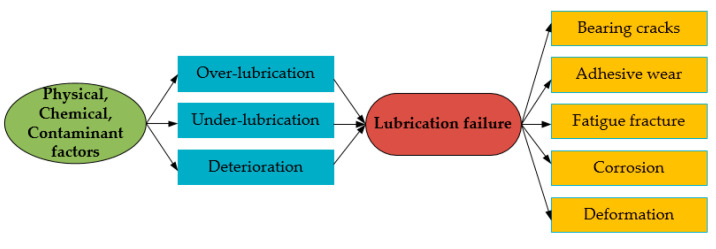
Analysis of lubrication failure of wind power bearings [70,109].

**Figure 13 polymers-14-03041-f013:**
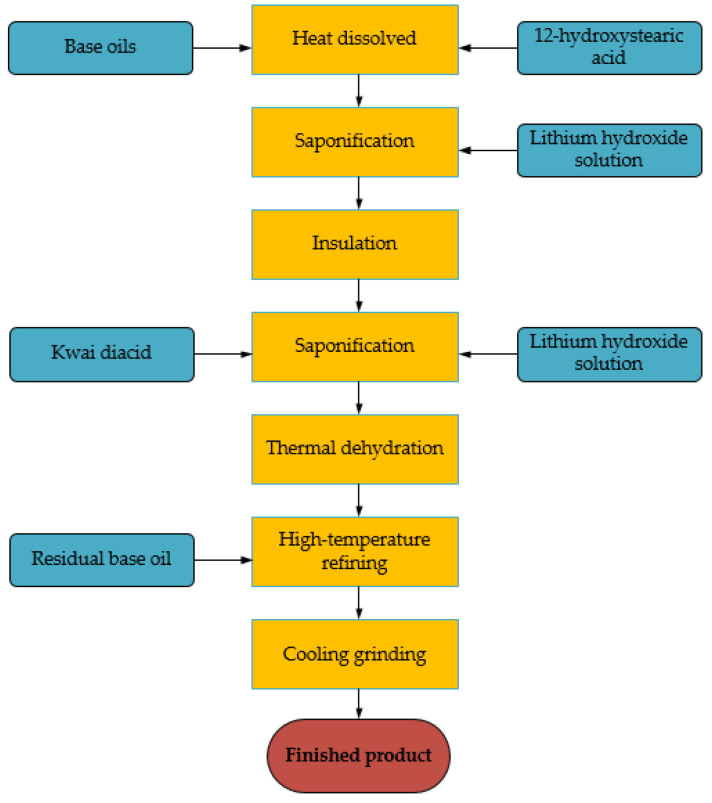
Flow chart of preparing wind turbine bearing grease (Adapted from [130]).

**Figure 14 polymers-14-03041-f014:**
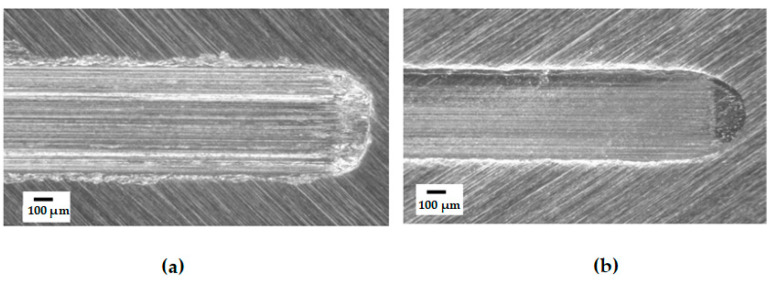
Optical micrographs of worn surfaces after a test (1.5 Hz, 454 m, 45 N), (**a**) Lubricated with PAO; (**b**) 5 wt.% IL in PAO (Adapted from [140]).

**Figure 15 polymers-14-03041-f015:**
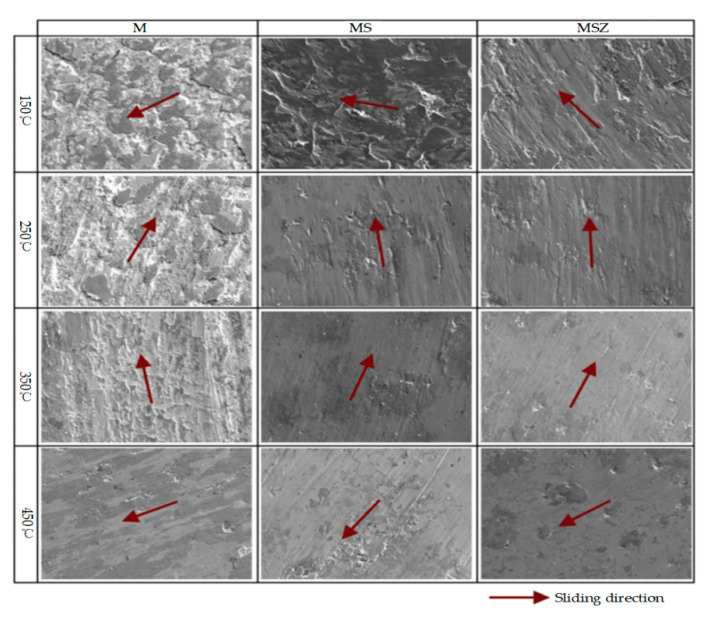
EPMA of frictional surface of M, MS, and MSZ under different temperatures (Reprinted with permission from Ref. [142]. Copyright 2020 Elsevier).

**Figure 16 polymers-14-03041-f016:**
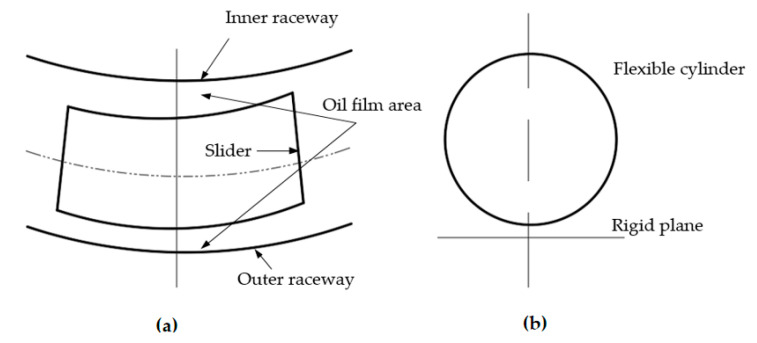
Lubrication structure of slider, (**a**) Lubrication area; (**b**) Cylinder contact with rigid. (Adapted from [167]).

**Figure 17 polymers-14-03041-f017:**
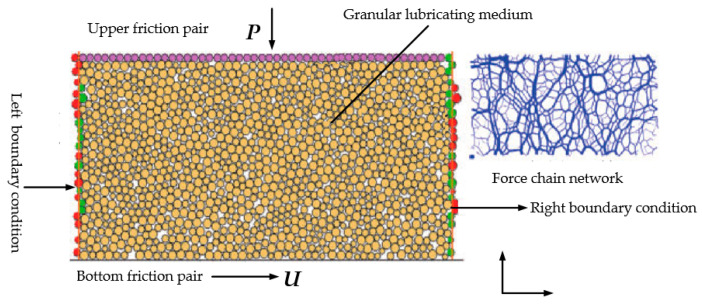
Discrete element model for particle flow lubrication (Adapted from [169]).

**Figure 18 polymers-14-03041-f018:**
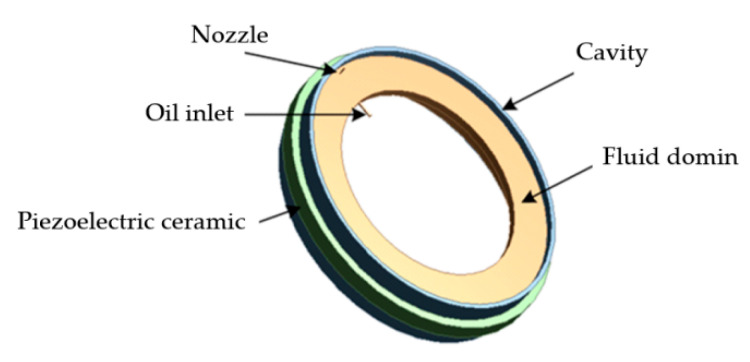
Finite element model of the piezoelectric microjet (Adapted from [172]).

**Figure 19 polymers-14-03041-f019:**
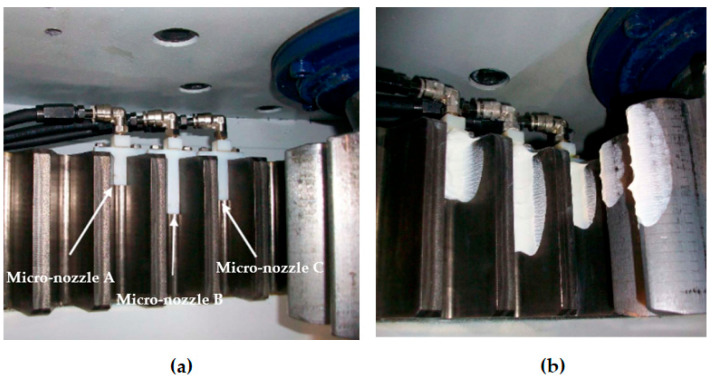
Micro-nozzle integration in the pitch gear, (**a**) 2, 4, and 3-cm length micro-nozzle; (**b**) Grease distribution on the tooth flank (Reprinted with permission from Ref. [173]. Copyright 2017 Springer).

**Figure 20 polymers-14-03041-f020:**
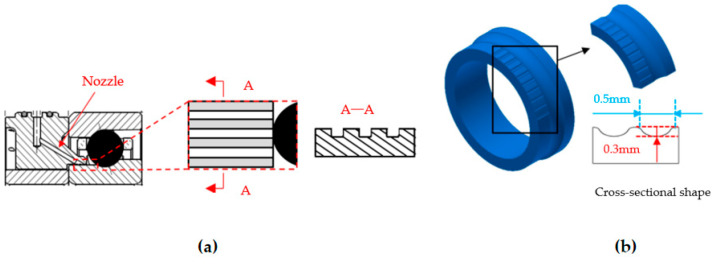
Axial groove structure and design position in the bearing, (**a**) Axial groove structure; (**b**) Bearing inner ring with groove structure (Adapted from [175]).

**Figure 21 polymers-14-03041-f021:**
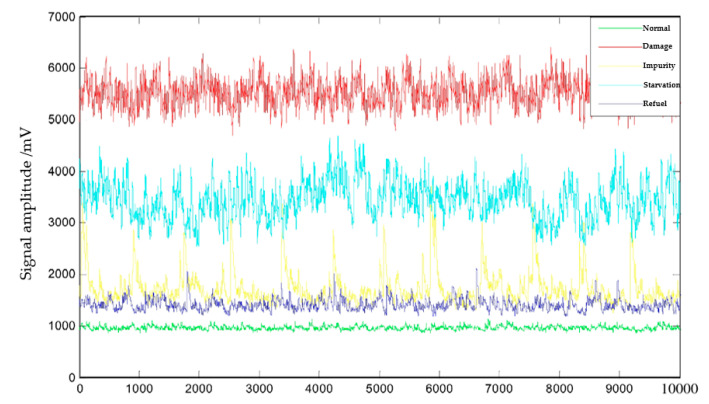
Waveforms of ultrasonic signals on different operation conditions (Adapted from [184]).

**Figure 22 polymers-14-03041-f022:**
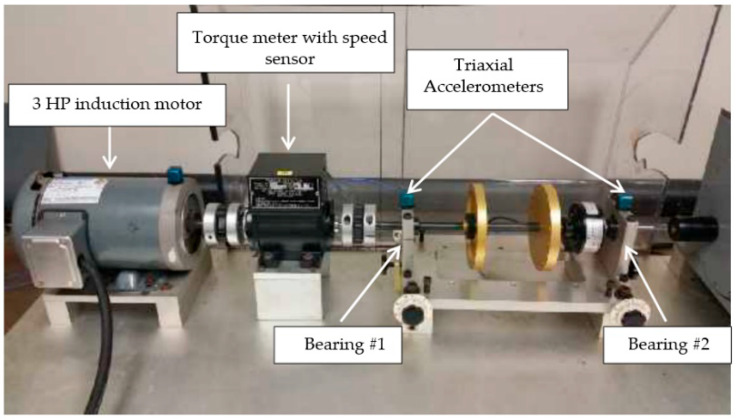
Test benches for experimentation (Adapted from [189]).

**Figure 23 polymers-14-03041-f023:**
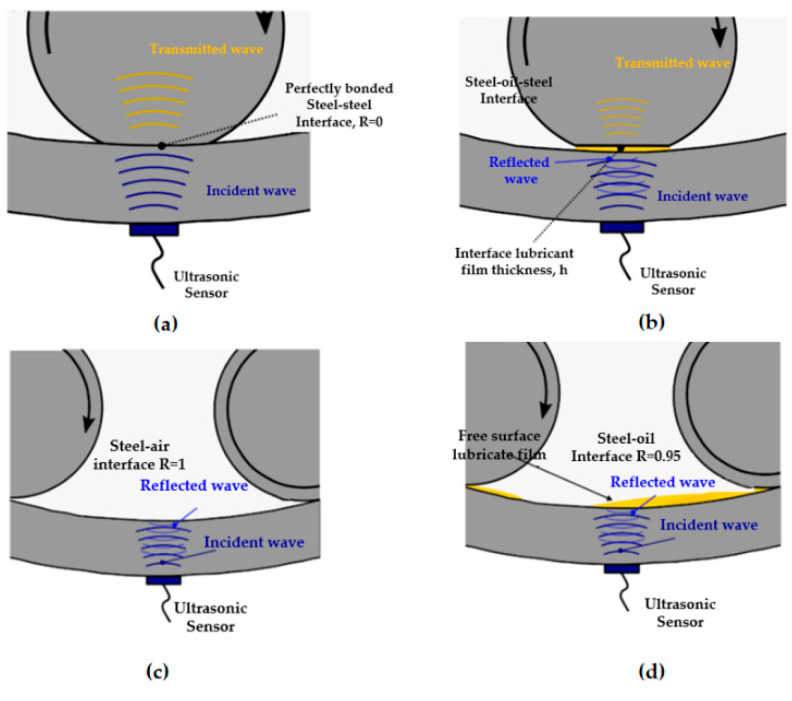
Conceptual diagram of ultrasonic reflectometry [190]. (**a**) Steel–steel interface; (**b**) steel–oil–steel interface; (**c**) steel–air interface; (**d**) steel–oil interface.

**Figure 24 polymers-14-03041-f024:**
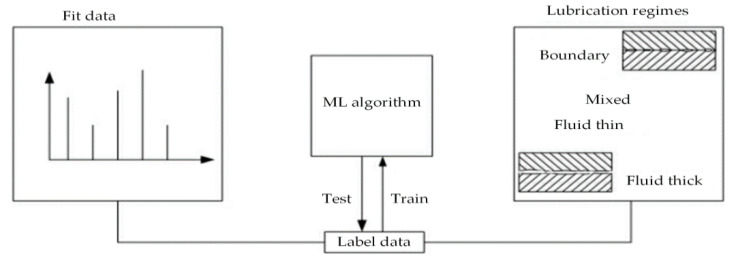
Approach for the classification of lubrication regimes [199].

**Figure 25 polymers-14-03041-f025:**
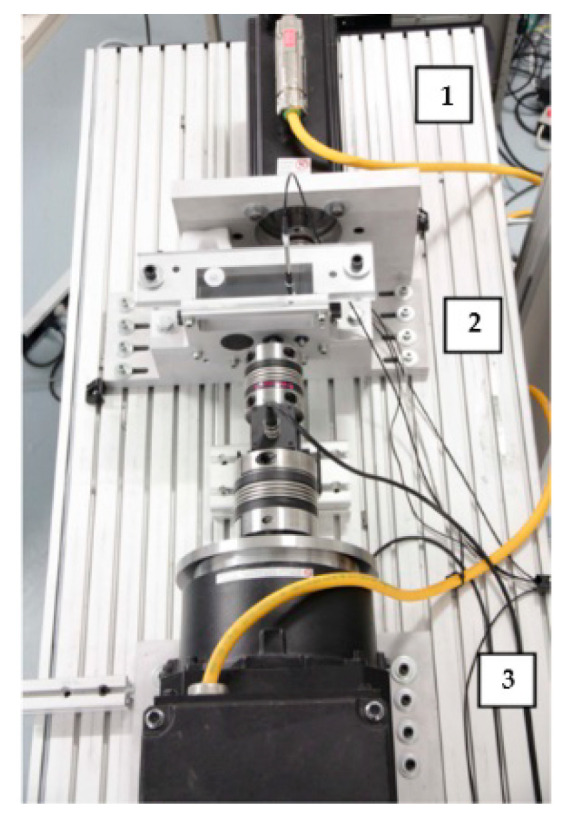
Illustration of the experimental setup, (1) Drive motor; (2) Lubrication gear; (3) Load motor (Reprinted with permission from Ref. [202]. Copyright 2012 Springer).

**Figure 26 polymers-14-03041-f026:**
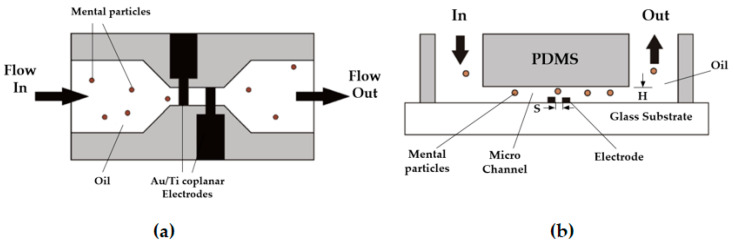
Schematic picture of the microfluidic sensor, (**a**) Top view; (**b**) Front view (Reprinted with permission from Ref. [203]. Copyright IOP Publishing).

**Figure 27 polymers-14-03041-f027:**
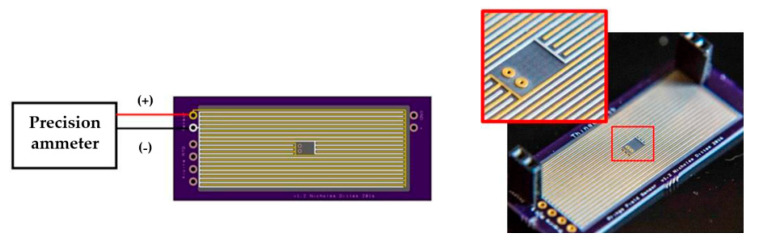
Sensor figures (ENIG and zinc coating indicated by gold and white traces) (Reprinted with permission from Ref. [204]. Copyright 2020 Taylor & Francis Group, LLC).

**Figure 28 polymers-14-03041-f028:**
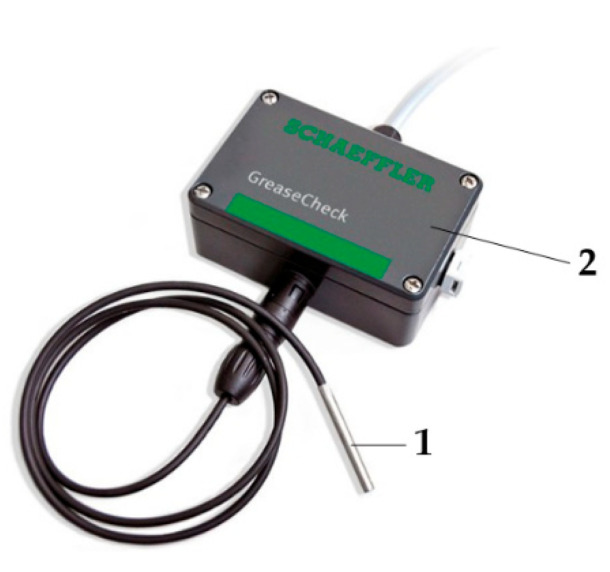
Grease sensor and electronic evaluation system, (1) Grease sensor; (2) Electronic evaluation system (Adapted from [205]).

**Figure 29 polymers-14-03041-f029:**
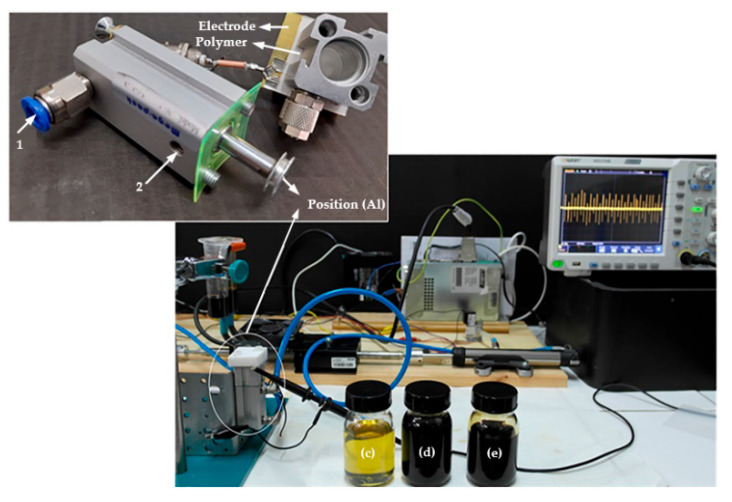
A pneumatically driven online triboelectric oil oxidation sensor (Reprinted with permission from Ref. [212]. Copyright 2022 Elsevier).

**Figure 30 polymers-14-03041-f030:**
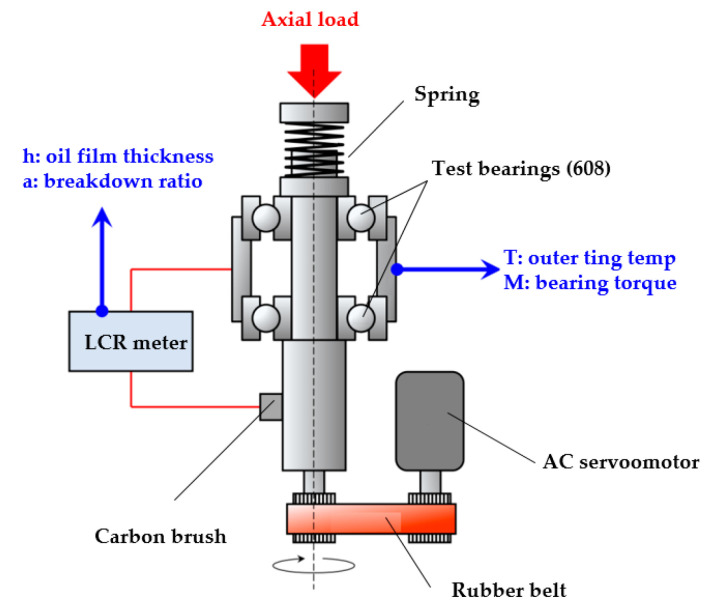
Bearing test rig for electrical impedance method [213].

**Table 1 polymers-14-03041-t001:** Common types of wind turbines [19,20].

Basis	Category Name
Spindle to ground Position	Horizontal, vertical.
Generator types	Synchronous, asynchronous.
Loading mode	Lifting, resistance.
Operation mode	Constant-speed, limited-variable-speed, variable-speed.
Power	Micro, small, medium, large.
Grid connection	Grid-connected, off-grid.
Drive train	Gearbox speed increase, direct drive.
Blade number	Multi-blade, three-blade, double-blade, single-blade.

**Table 2 polymers-14-03041-t002:** Types of bearings for different sections (Adapted from [11,52]).

Bearing Type	Components Used
Cylindrical, Spherical, Tapered roller bearings	Main shaft, Gearbox
Cylindrical and Spherical rolling element bearings	Generator
Single/Double row contact ball bearings	Pitch/Yaw, Gearbox
Cylindrical roller bearings	Main shaft, Gearbox, Generator

**Table 3 polymers-14-03041-t003:** Frictional failure analysis of wind turbine bearings (Adapted from [67,71,72]).

Failure Type	Failure Modes and Characteristics	Failure Cause
Fatigue failure	Characteristics: as shown in Figure 4a, bearing surface appears to be peeling and other phenomena. Modes: white microstructure flaking, micropitting, and axial cracking.	Manufacture technology issues, long-term high-load operation, bearing rolling contact fatigue.
Wear	Characteristics: as shown in Figure 4b, bearing clearance increases and wear marks appear, which are often accompanied by abnormal vibration and noise, and the color of the bearing becomes darker. Modes: bearings are prone to adhesive wear, abrasive wear, Brinell marks, and corrosion patterns.	Bearing internal debris roller or raceway internal debris cause surface damage or lead to the appearance of chips and dents.
Corrosion	Characteristics: as shown in Figure 4c, rust spots and shallow pits on the outer surface of the bearings, with grey and black spots along raceways. Modes: the chemical reaction caused by the bearing surface material is divided into physical corrosion and chemical corrosion.	Physical and chemical corrosion caused by debris and water or corrosive substances in the bearings.
Electrico-etching	Characteristics: as shown in Figure 4d, rust on the outer surface of the bearing and on the raceways and rollers, accompanied by shallow pits and dark grooves. Modes: there is rust on the outer surface of the bearing and on the raceways and rollers.	When the axial current exists or the voltage is too high, the axial current flows through the rolling body and raceway, causing sparks and arcs, thus melting the bearing surface.
Plastic deformation	Characteristics: as shown in Figure 4e, the bearing appears loose, and the surface becomes twisted and deformed. Modes: Bearings material appears indentation bending and other phenomena.	Plastic deformation comes from misaligned motion, and there are two different types. The first type is on the macroscopic scale, caused by heavy or high loads with excessive accelerating loads, and occurs at the contact surface of the contact ball race. The second type is on the microscale and occurs in a small indentation in the contact surface. Misalignment of raceway edge and ball path occurs or relative motion of bearing ring and shaft.
Cracks and fractures	Characteristics: as shown in Figure 4f, cracks or fissures on the bearing surface. Modes: stress fracture, fatigue fracture and thermal cracking modes.	Excessive stress concentration in the tensile strength of the bearing material, high temperature caused by sliding friction and improper assembly or process problems.

**Table 4 polymers-14-03041-t004:** Lubricants for wind turbine bearings (Adapted from [83,84]).

Bearing Type	Lubricants
Main shaft bearing	Grease
Generator bearing	Grease
Pitch Bearing	Grease
Yaw Bearing	Grease
Gearbox bearing	Gear oil

**Table 5 polymers-14-03041-t005:** Summary of the current status of wind power bearing lubricant research.

Reference Number	Author (s), Year	Major Findings
[129]	Sun et al., 2012	Synthesis of an excellent sulfur-containing heterocyclic derivative additive (RHY317).
[140]	Iglesias et al., 2015	Ionic liquids are used as additives to lubricants.
[144]	Ali et al., 2016	Different sizes of Al_2_O_3_ and TiO_2_ nanoparticles were used as lubricant additives to reduce friction.
[143]	Essa et al., 2017	The use of high content of mixed solid lubricants will greatly reduce the coefficient of friction.
[130]	Xia et al., 2018	A new type of wind power bearing grease (additive: T351, T323) was researched.
[146]	Haque et al., 2018	Lubricants containing metal additives are prone to water contamination and cause white etch cracks (WECs).
[141]	Ali et al., 2019	Solid lubricants have excellent friction reduction properties at high temperatures.
[131]	Mutyala et al., 2019	Proposed combination of solid lubricant two-dimensional MoS2 with diamond-like (DLC) film.
[142]	Elsheikh et al., 2020	Demonstrated that the combination of different solid nano-lubricants will have a positive effect on improving the tribological properties of bearing steel.
[134]	Gao et al., 2020	Developed a grease suitable for wind turbine spindle bearings.
[136]	Gao et al., 2020	Developed a new type of grease for wind turbine pitch bearings.
[137]	Schwack et al., 2020	Six industrial greases with different compositions were tested for their anti-wear performance.
[132]	Ng et al., 2021	Synthesis of a non-polluting and non-toxic nano-lubricant.
[138]	Sun et al., 2021	L1-320 extreme pressure-anti-wear performance is better than L2-320.
[139]	Saidi et al., 2021	Prepared MoS2 nano-lubricant additives.
[145]	Li et al., 2021	Extreme pressure additives and solid additives can cause corrosion on the surface of the bearing steel.
[147]	Li et al., 2021	Replace the lubricant for bearings whose drip point drops more than 30 °C and whose iron content exceeds 5000 ppm.
[148]	Feng et al., 2021	Reprocessing of wind turbine gearbox lubricants.
[133]	SKF Corp.	Developed about wind power main shaft bearing LGWM series grease.

**Table 6 polymers-14-03041-t006:** Common lubrication methods for wind turbine bearings [119,161,162,163].

Lubrication Method	Advantages	Disadvantages
Centralized lubrication	Lubrication pump centralized pumping lubricates, high efficiency, easy maintenance and overhaul, and high precision can achieve a variety of and self-quantitative lubricate supply [163].	The piping is complicated, the hydraulic pipe Louis leaks, the lubricant flow requirements are high, and the distributor is easily clogged causing system failure [119].
Oil mist lubrication	Better sealing effect on lubricant, minimizes waste of lubricant, less lubricant required, and less impurity intrusion.	Lubricants require high-level viscosity; the conveying distance is too short and the adjustment of the oil mist volume is difficult.
Oil–air lubrication	No requirement for lubricant viscosity, wide range of applications; the impact to the environment is lower; controllable lubrication quantity and high lubrication efficiency and high pressure airflow can cool down and resist impurities [161].	Need compressed air and other air sources; produce a certain amount of noise pollution; higher operational standard and total cost.
Oil bath lubrication	Simple lubrication structure; adequate lubrication; low operating cost.	Only suitable for low and medium-speed bearing lubrication. Oil quantity control is complicated and produces impurities, causing damage [162].

**Table 7 polymers-14-03041-t007:** Summary of research on wind turbine bearing lubrication methods.

Reference Number	Author (s), Year	Major Findings
[176]	Erill et al., 2013	Invented a dynamic lubrication system for wind turbine pitch bearings.
[164]	Yao et al., 2015	Developed a new type of lubrication pumping station.
[170]	Westerberg et al., 2016	It shows that the elbow angle of the nozzle is related to the flow of grease, and the high flow rate will reduce the sliding effect.
[165]	Wang et al., 2016	Theoretical analysis model of wind power bearing lubrication state was established.
[177]	Yang et al., 2016	Achievement of dynamic lubrication of wind turbine main shaft bearings
[173]	Farré-Lladós et al., 2017	Designed and manufactured micro-nozzles for automatic lubrication of wind turbine pitch gears.
[167]	Gu et al., 2019	Structural design of rolling bearings was carried out.
[168]	Peng et al., 2020	Design of automatic temperature-controlled heating device.
[171]	Liu et al., 2021	A fluid–solid coupling simulation model was developed to study the effects of oil flow rate, lubricant viscosity, nozzle angle, and nozzle number on the lubrication characteristics of gearbox bearings.
[169]	Meng et al., 2021	Constructed the discrete element numerical model of particle flow lubrication.
[172]	Li et al., 2021	Designed a bearing lubrication device based on piezoelectric micro-jet theory.
[174]	Oliveira et al., 2021	A new method for modeling and identifying the oil supply in dynamic pressure bearings is proposed.
[175]	Ge et al., 2021	Proposed to add a groove structure to the non-contact area of the bearing inner ring surface.

**Table 8 polymers-14-03041-t008:** Summary of research on wind turbine bearing lubrication monitoring methods.

Reference Number	Author (s), Year	Major Findings
[210]	Raadnui et al., 2005	Developed a low-cost gate capacitive sensor.
[203]	Murali et al., 2009	A pico-fluidic apparatus was designed for the inspection of metals particulates in low conductivity lubricants.
[198]	Ünlü et al., 2012	Determined the frictional wear behavior under bearing lubrication conditions.
[202]	Márton et al., 2012	Proposed an energy-balanced lubricant viscosity detection method.
[184]	Su et al., 2013	Used the signal amplitude graph reflects the lubrication status of the bearing.
[191]	Ibrahim et al., 2017	Used acoustic emission technology to test solid contaminants of different concentrations and particle sizes of grease.
[199]	Moder et al., 2018	Using machine learning (ML) algorithms to predict the lubrication status of high-speed torque bearings.
[189]	Mijares et al., 2019	Presented a vibration analysis method to monitor the amount of lubrication in the bearing.
[213]	Maruyama et al., 2019	Developed an electrical impedance method for monitoring the lubrication status of deep groove ball bearings.
[204]	Dittes et al., 2020	Developed a corrosion sensor.
[201]	Katsaros et al., 2021	Combining numerical analysis and machine learning techniques.
[190]	Nicholas et al., 2021	Suggested a new ultrasound reflection technique.
[200]	Zavos et al., 2022	Developed an ML model based on an ordered regression algorithm to predict coating application and lubricant selection for bearing components.
[212]	Karluk et al., 2022	Proposed a method for monitoring lubricant degradation based on frictional charging.
[205]	Schaeffler Co.	Created a device for grease condition monitoring.

## Data Availability

Not applicable.

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
