# Peer review of "Review of Tribological Failure Analysis and Lubrication Technology Research of Wind Power Bearings"

_polymers, 2022, doi:10.3390/polym14153041_

Round 1

Reviewer 1 Report

The manuscript is very clear and concisely presented in terms of contents and vocabulary. The manuscript is well written but there are some enhancements that need to be added to the manuscript (Major Revisions). The following points are major issues:

1.   In the Introduction, the gap of research work (scientific problem) is neither adequately elaborated. Include statements on the significance of the review for saving energy at the end of the introduction section.

2.   The wear and friction mechanisms were not well discussed by SEM and EDS figures. Please, explain the responsible mechanisms clearly. This is one of the key discussions that this manuscript has to have in the results and discussion to show the novelty of the study.

3.   The literature seems to be weak in terms of the publication survey and needs to be enhanced including some of the recent publications on the use of self-lubricating of materials and nanolubricants. Herein, some recent publications are given below as examples.

https://doi.org/10.1007/s11249-019-1183-6

https://doi.org/10.1177/1350650117723224

https://doi.org/10.1016/j.triboint.2016.08.011

4.   The review articles are an analysis of the topic rather than simply a collection of information taken from different publications. Please provide analysis in terms of what is the current status of the field, what are the challenges.

5.   The authors may add a section of "Future Perspective and Recommendations" after the conclusions.

Author Response

We would like to thank the reviewer for taking out your invaluable time to thoroughly comment on our submitted work and revision. Please see the attachment.

Reviewer 2 Report

The authors present an extensive overview of bearings lubrication technologies and tribological failures in wind turbine applications. These types of review articles shed light on the state-of-the-art related to this field and can promote follow-up research.

The manuscript is quite clear and well-written. On this occasion I only have some minor remarks:

1. This review article seems more suitable for a Tribology journal than for Polymers. Why did you select this journal instead of any Tribology journal?

2. Machine learning approaches have been used to estimate the bearing failure stages of wind turbine gearboxes, e.g., Elasha et al. (DOI: 10.3390/s19143092). The manuscript could be improved by adding previous works dealing with machine learning based prognosis.

3. Concerning the possible use of micro-nozzles, like those in reference [167], the grease flow through elbow channels was studied by Westerberg et al. (DOI: 10.1007/s11249-015-0469-6). Maybe the authors would like to discuss this aspect further by considering this new reference or other similar works.

Round 2

Reviewer 1 Report

The authors have considered and incorporated all the major changes that I requested. Hence, it is recommended for publication.